# Fostering Sustainable Fashion Innovation: Insights from Ideation Tool Development and Co-Creation Workshops

**Eunsuk Hur [1],\* and Katie Beverley [2]**

1   Faculty of Arts, Humanities and Cultures, School of Design, University of Leeds, Leeds LS2 9JT, UK
2   PDR, The International Centre for Design & Research, Cardiff Metropolitan University, Western Avenue, Cardiff CF5 2YB, UK; kbeverley@cardiffmet.ac.uk
\*   Correspondence: e.s.hur@leeds.ac.uk

**Abstract:** Idea generation is often considered the biggest influence on both the value creation and sustainability of a product–service system. Although several researchers have put forward sustainable innovation tools, there has been limited research into potential tools that can support the ideation stage for future sustainable fashion professionals. In this study, we aim to address this gap by (1) critically examining the management control systems that support the ideation phase and how different types of tools assist sustainability innovation, (2) investigating the potential value of co-creation in the ideation phase, and (3) evaluating a sustainable fashion toolkit designed for use in co-creation workshops. The proposed application of the toolkit was tested with designers, marketers, and entrepreneurs in several co-creation workshops (*n* = 147) that examined the users' experiences and the toolkit's usefulness using task analysis. In-depth interviews with industry professionals and educators were also conducted to identify the key criteria for optimal tool development and use in both industrial and educational contexts. This study contributes at a theoretical level by proposing a sustainable fashion innovation tool that considers management control systems and practical guidelines for tool development and by delineating implications for the future of sustainable fashion education and skills.

**Keywords:** sustainable fashion tools; co-design; design thinking; product service systems (PSS); management control systems (MCS)

## 1. Introduction

The textile and apparel industry is staggering under the weight of its environmental and social burdens. It is responsible for an estimated 8–10% of global carbon emissions, 20% of industrial water, and 35% of ocean microplastic pollution, consumes prodigious quantities of resources (including water), and generates similarly vast amounts of waste [1]. Meanwhile, despite some excellent NGO- and industry-led initiatives in recent years that have increased transparency, as of 2018, garments were second only to consumer electronics as the global exports most likely to feature modern slavery in their supply chains [2]. Child, forced, and bonded labour have been reported [3,4], most recently amongst the Uighur population working in the Xinjiang cotton supply chain [5]. Whilst most of this impact is manifest in low- and middle-income countries, reports of abusive labour practices amongst suppliers in the UK [6] have been substantiated during the recent COVID-19 pandemic [7], suggesting that even high-income countries with stringent health, safety and labour legislation are not immune.

Such sustainability concerns have led to increasing scrutiny from intergovernmental organisations (e.g., the United Nations Alliance for Sustainable Fashion) [8], governments (e.g., the UK's Environmental Audit Committee) [6], NGOs (e.g., Labour Behind the Label; Fashion Revolution; Greenpeace) [9–11], supply chain partners [12], and consumers. Pressure from stakeholders has long been recognised as a significant driver of the adoption

of sustainability practices [13]. Recent academic perspectives propose that while external stakeholder pressure is crucial for building the business case for sustainability adoption, internal stakeholders, including managers and employees, play a pivotal role in transforming businesses from making reactive responses to sustainability threats to effecting more proactive or deep-seated improvements in operational performance [14,15]. The acquisition, distribution, interpretation, and commitment to the organisational memory of knowledge related to sustainability amongst internal stakeholders are necessary for firms to create the most value from sustainability initiatives.

Past studies [16–18] have also pointed to the potential for co-design to leverage sustainability practices by promoting stakeholder engagement, and facilitating debate and collaboration with and among them, in the process achieving alternative sustainable solutions. The key benefits include (1) creating mutually valuable product and service ideas [19,20], (2) strengthening the personal and user experience value by promoting active participation in the design process [21], (3) refining project management and consumer communication strategies, and (4 and 5) supporting the positive social value by improving users' quality of life [20] and the accompanying commercial value by improving long-term brand loyalty [21]. A co-design approach can allow stakeholders to act as agents of change in the transformation process, rather than being subjected to changes imposed from the outside, aligning the transformation outcomes with the community's interest and benefit [22].

Although incorporating sustainability via co-design has the potential to deliver this value, it is important to acknowledge its weaknesses. For instance, Dubois et al., argue that co-design has often lacked a common purpose [23], and that addressing critical issues or exploring the potential of innovative solutions through co-design can be time-consuming. In the fashion sector, some sustainability tools and approaches have been introduced, such as Sustainable Fashion Bridges (SFB) [24], TED's TEN [25], and the reDesign canvas [26]. Nevertheless, the mechanics of how to effectively integrate sustainability information in the new product–service system (PSS) design, and specifically during the crucial ideation process, are still not well understood. Even if they do address sustainability from the outset, such tools for designers or fashion entrepreneurs have often supplied too little guidance on how to effectively utilise them in the co-creation process, and empirical studies often fall short of addressing this question or do so only to a limited degree.

In this study, we sought to address the following research questions: (1) What types of tools are needed to effectively support the development of sustainable fashion innovation? (2) How well do existing tools meet these needs? (3) What role could a card-based toolkit intended to support designers, entrepreneurs, marketers, and other stakeholders do to effectively co-create sustainable fashion design strategies? (4) When developing a sustainable fashion tool, what key criteria enable its effective use in higher education and the fashion industry?

To address these questions, this study conducted a critical review of the literature on the new product development (NPD) process, with specific emphasis on the skills and competencies that future fashion sustainability professionals need during the ideation stage. The study also examined the existing literature on sustainable fashion tools that target any point in the product lifecycle. We then tested the application of a sustainable fashion tool (SFB) in the setting of co-design workshops and reviewed how effectively the tool meets question 4′s criteria. Finally, we made recommendations for underpinning future research directions regarding the SFB by identifying preferences for sustainability tools in the industry and education sectors and methods of serving future sustainable fashion professionals. This study contributes to the development of a framework for co-design workshops in sustainable fashion innovation and supports the creation of novel product–service system designs that align with sustainable fashion innovations.

## 2. Literature Review

### 2.1. The Design Process in the Fashion Industry

NPD in the fashion industry is often chaotic in nature, with roles and duties being frequently unclear [27]. Within a typical mass-market fashion business, the NPD process is a multidisciplinary team effort, with each member providing different forms of input to the innovation process [28–31]. Technology-driven innovation in materials, fibres, and processes are the domain of textile and garment technologists; supply chain innovation is in the province of procurement and logistics experts; and business model innovations are strategic decisions usually requiring leadership at the senior management level [32]. Meanwhile, fashion designers are often responsible for 'stylistic innovation', which Cappetta et al., describe as 'the change of the aesthetic characteristics of a product, generating both a new product—from a physical point of view—and a new meaning' [33]. The fashion design function usually consists of a small team under the leadership of a lead designer who is responsible for setting the context for the innovation in meaning and managing the creative process [29].

In his study on an Italian fashion design company, Cirella has conceptualised multidisciplinary teams responsible for fashion product development as micro-social systems motivated to co-operate to achieve a common goal through the application of collective creativity [30]. Interactions between members of the micro-social system can influence the effectiveness of collective creativity. Hargadon and Bechky have identified four types of social interaction important for collective creativity: (1) help seeking, in which team members actively request the assistance of others; (2) help giving, where team members offer unsolicited assistance to others; (3) reflective reframing, in which team members reflect respectfully upon and build on the comments and actions of other team members; and (4) reinforcement, in which the informal and formal management of help seeking, help giving, and reflective reframing ensures those interactions' continuation in the group [34].

Systems that support positive social interactions have the benefit of democratising the knowledge held by the team. This is important in the context of sustainable fashion, since junior designers are more likely to have explored sustainability in their fashion courses and may have the highest level of sustainability-awareness in the team [35–38]. Davila and Ditillo have identified the importance of management control systems (MCS) in managing the collective creative process in fashion design [29]. They describe two different types of MCS: directional systems 'define the rules of the game', providing structural boundaries for the micro-social system, while inspirational systems support productive social interactions within it. Examples of directional systems include cost cards that report the costs of creative proposals and provide an interface between the lead designer and the marketing function, sales reports that inform designers and buyers about market acceptance of different styles, and collection briefs that are supplied from the marketing department and define the number and type of items in the collection. Inspirational systems include research trips in which designers visit inspirational places such as fashion capitals, fashion shows, and industry trade shows; theme and mood boards developed by the lead designer that provide a common vision for the collection; and collages that provide common visual references related to the overall theme. These inspirational activities are the systems that support the divergent ideation phase of the design process.

### 2.2. The Provision of Sustainability Information as MCS

Processes that are employed to integrate sustainability in the fashion design process may be viewed as forms of management control systems. In their recent study on sustainable fashion design strategies in industry, Claxton and Kent found that sustainability information is usually supplied to fashion designers from technologists, buyers, and other sustainability specialists [28]. Industry-developed measurement and ranking tools such as the Nike Maker App [39] and the Higg Index [40] have also garnered interest across the industry [41]. In both cases, the role of sustainability information is to align the creative outcome with the needs of the other functions. In other words, sustainability information

constrains, rather than inspires, the collective process and may be considered to act as a directional MCS.

Directional MCSs aimed at supporting sustainability play an important role; sustainable concepts are more likely to be adopted if they have been informed by market, technical, and cost considerations [42]. However, ranking systems tend to focus on cradle-to-gate activities that can be more easily monitored and measured and adopt a cursory approach to quantifying and mitigating consumer-related impacts. As such, they do not necessarily support the generation of novel design concepts that utilise style innovation to create new sustainable meanings for consumers (for example, designs that promote emotional engagement and longevity, encourage customisation, use zero-waste construction methods to create a new aesthetic, promote more sustainable care regimes, and empower consumers to make better purchase decisions).

Without accompanying inspirational MCS, directional systems might also inhibit the creative process. Collado-Ruiz and Ostad-Ahmad-Ghorabi explored the influence of directional environmental information on the creativity of a group of design engineers [43]. They found that the more prescriptive the sustainability information, the more conventional the resulting solutions. As Deutz et al., comment in their analysis of ecodesign tools: "sustainability criteria...imposed on the design process as a limiting factor/design criteria...expressly do not aid in the generation of concepts and, consequently, if the divergent stage of design has not been performed well, then choices are likely being made between sub-optimal alternatives" [44].

### 2.3. Design for Sustainable Fashion Innovation

The concept of sustainable innovation is commonly employed in conjunction with the term 'eco-innovation' and has its roots in broader principles, including 'environmental sustainability' and 'sustainable development' [45]. Building on the concepts of 'eco-innovation' and 'sustainable development', the definition of sustainable innovation can have multiple meanings and qualities in various contexts because these requirements shift depending on their time, place, and social integration [46]. The shift from 'green' to 'eco' to 'sustainable' design reflects a broadening of theoretical and practical application, as well as, to some extent, an intensifying critical viewpoint on ecology and design [47]. One of the widely accepted definitions of 'sustainable innovation' is 'a process where sustainability considerations (environmental, social, and financial) are integrated into company systems from idea generation through to research and development (R&D) and commercialisation. This applies to products, services, and technologies, as well as to new business and organisational models [46,48]. This definition attempts to address the social and ethical aspects of sustainability in addition to the economic and environmental aspects.

In the context of sustainability, the five dimensions of sustainable fashion innovation were emphasised by Kozlowski et al. They include the traditional triple bottom line of sustainability—environmental, social, and economic—along with two additional dimensions: aesthetic and cultural. While cultural sustainability necessitates a systems-level approach, the aesthetic dimension manifests at the product level [49]. The Kozlowski et al., definition emphasises that the design process can evolve into a more forward-thinking, collaborative process that promotes sustainability and visual appeal [49]. This requires identifying the numerous interconnected, complex issues that should be considered in sustainable design. However, fashion designers new to sustainable design may find it overwhelming and challenging to comprehend the complicated nature of sustainability issues in the fashion system [41]. To address these challenges, some scholars and organisations have offered different types of tools that address sustainability innovation. Depending on their scope, tools can be divided into four levels [50]. Tools at Levels 1 and 2 frequently serve as the catalyst for incremental innovation, which results in some improvement in the company's design expertise in its existing field [51]. Tools at Levels 3 and 4 enable more radical innovation, may disrupt established procedures or concepts and may require more radical transformations or the adaption of cutting-edge technologies [52].

Level 1 tools are most common in the fashion and textile industries. They typically use matrices and guidelines for lifecycle analysis (LCA). LCA has the advantage of being able to assess the possible environmental effects of the materials or products used in product development, but because it requires more information, LCA can be challenging to employ during the early design stages (design brief, strategy formulation, and concept design) [17]. Level 2 tools often relate NPD to other companies' strategies, including those involving their business, marketing, manufacturing, and sustainable management systems. Such Level 2 tools can expose weaknesses in a company's strategy regarding the environmental aspects of its product's lifecycle. A well-known comparative tool is the Lifecycle Design Strategy (LiDS) wheel, which provides a general overview of products' potential to improve environmental effects [51].

Level 3 tools enable the creation of new product features or service concepts by integrating product development procedures into the management of the product supply chain. Engaging a larger variety of stakeholders and incorporating their values into the company's model constitutes a significant leverage point for sustainability [53].

Level 4 tools commonly concentrate on industry- or society-wide associations [50]. This industrial–ecology strategy entails businesses, organisations, and communities co-creating sustainable innovation to promote wider social and environmental benefits.

Most tools currently used in the fashion and textile sectors are Levels 1 and 2 tools. Although these tools are helpful for identifying environmental risks and impacts across products, value chains, and facilities, these LCA-related tools often require larger quantities of data, making them challenging to use in the ideation process due to time and financial constraints, especially for small-sized fashion businesses or in the context of educating and supporting sustainable fashion professionals.

Whilst industrially developed tools tend to constrain the design process, academic interest has increasingly focused on supporting sustainable ideation in fashion design [41]. Kozlowski et al. [26] proposed a sustainable business model tool for fashion entrepreneurs that is a redesign of Osterwalder and Pigneur's business model canvas [54]. The original canvas contains nine building blocks: value propositions, customer segments, customer relationships, key activities, key partners, key resources, channels, cost structures, and revenue streams. It is one of the tools most used by practitioners and researchers to evaluate an existing business model or to establish a new one. The redesigned canvas consists of 12 elements because it includes more detailed descriptions of design and material selection and product development in the context of sustainable fashion. This redesigned canvas offers a well-informed directional MCS and a useful holistic overview of business components for Level 2 innovation. Nevertheless, it offers limited support for strengthening the value proposition and striking off in new creative directions.

When introduced into the collective design process, ideation tools are, by their very nature, inspirational MCSs. Although they may take various forms, card-based ideation tools are a popular format. Examples of card-based tools for supporting sustainable fashion and textile design are TED's TEN [25] and Sustainable Fashion Bridges (SFB) [24]. TED's TEN is designed primarily for textile designers and offers practical guidance and inspiration for developing 'practice-based sustainable design strategies' based on card-based tools that aim to reduce negative environmental effects. It offers hands-on, practical, creative solutions for developing sustainable textile designs [25]. The scope of the TED's TEN is primarily to support textile designers; therefore, guidance on how interdisciplinary designers and fashion entrepreneurs can develop Level 2–4 sustainable fashion innovations through sustainable business models or industrial–ecology strategies is limited. In contrast, SFB provides a holistic perspective through which to address both fashion-system production and consumption processes and considers product lifecycles and interventions in the industrial ecosystem for Level 3–4 sustainable innovations [24,55]. Although the SFB tool is useful for the potential development of sustainable innovations in Levels 2–4, both the TED's TEN and SFB card-based tools require a highly skilled design facilitator to be used

effectively and lack both evaluation from users' perspectives and empirical studies on their effective use.

In their recent review and analysis of 155 card decks developed to support different aspects of design, Roy and Warren found that the majority of decks were designed to facilitate creativity in problem solving, often in specific domains, such as sustainability [56]. The authors concluded that, although evaluations of tools tend to be conducted by their developers in educational environments and are not, therefore, fully objective or contextual, well-designed card-based tools encourage interaction, offer different perspectives on design problems, and support structured design discussions that facilitate communication and develop shared understanding. As such, card-based design tools support the positive social interactions that Hargadon and Bechky propose as supporting collective creativity [34]. However, criticisms of card decks include that they include too much information, thereby inhibiting the creative process [43], oversimply information [56], and are susceptible to becoming dated as knowledge in the field evolves. This suggests that card-based tools may not always fulfil their intended role in supporting divergent aspects of the design process. After considering both the advantages and limitations of card-based tools, this study used the SFB toolkit to further investigate how card-based sustainable fashion tools can effectively operate as an inspirational MCS. The study also examined which other media could help fashion students, practitioners, and educators develop sustainable fashion innovation strategies and which other skills are required by future sustainable fashion professionals.

### 2.4. Sustainable Fashion Bridges

2.4.1. Toolkit Vision and Structure

Sustainable Fashion Bridges are a card-based design toolkit that supports fashion designers in considering the fashion system at personal, social, and environmental levels. Its structure was influenced by the Design with Intent Method [57] and the Human-Centred Design Toolkit by IDEO [58]. It consists of a set of 60 cards grouped according to six categories: Choice, Optimisation, Social Conversation, Persuasion, Empowerment, and Interaction (see Figure 1).

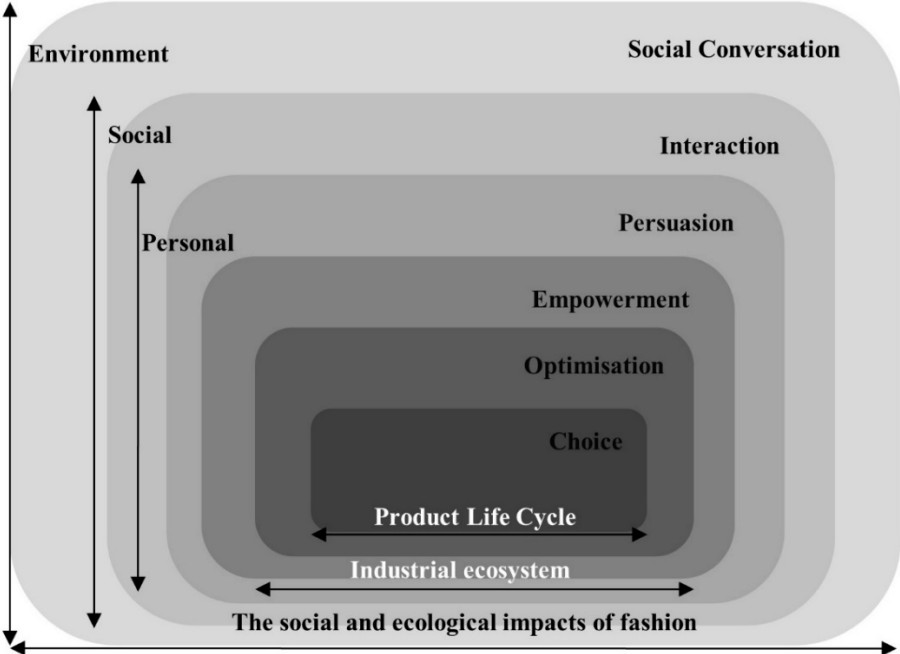

**Figure 1.** The SFB ideation toolkit framework.

The different categories and their combination allow designers to undertake ideation at multiple levels. 'Choice' explores individual product life cycles; 'Optimisation' considers interventions at the industrial ecosystem level; and 'Empowerment', 'Persuasion', 'Interaction', and 'Social Conversation' operate at the system level, encouraging connections between sustainable consumption and design for behaviour change. As Kozlowski et al., has discussed, SFB, therefore, includes strategies that have the potential to improve, challenge, and transform the existing fashion system [26,41]. Figure 2 shows the physical configuration of the toolkit and examples of the cards.

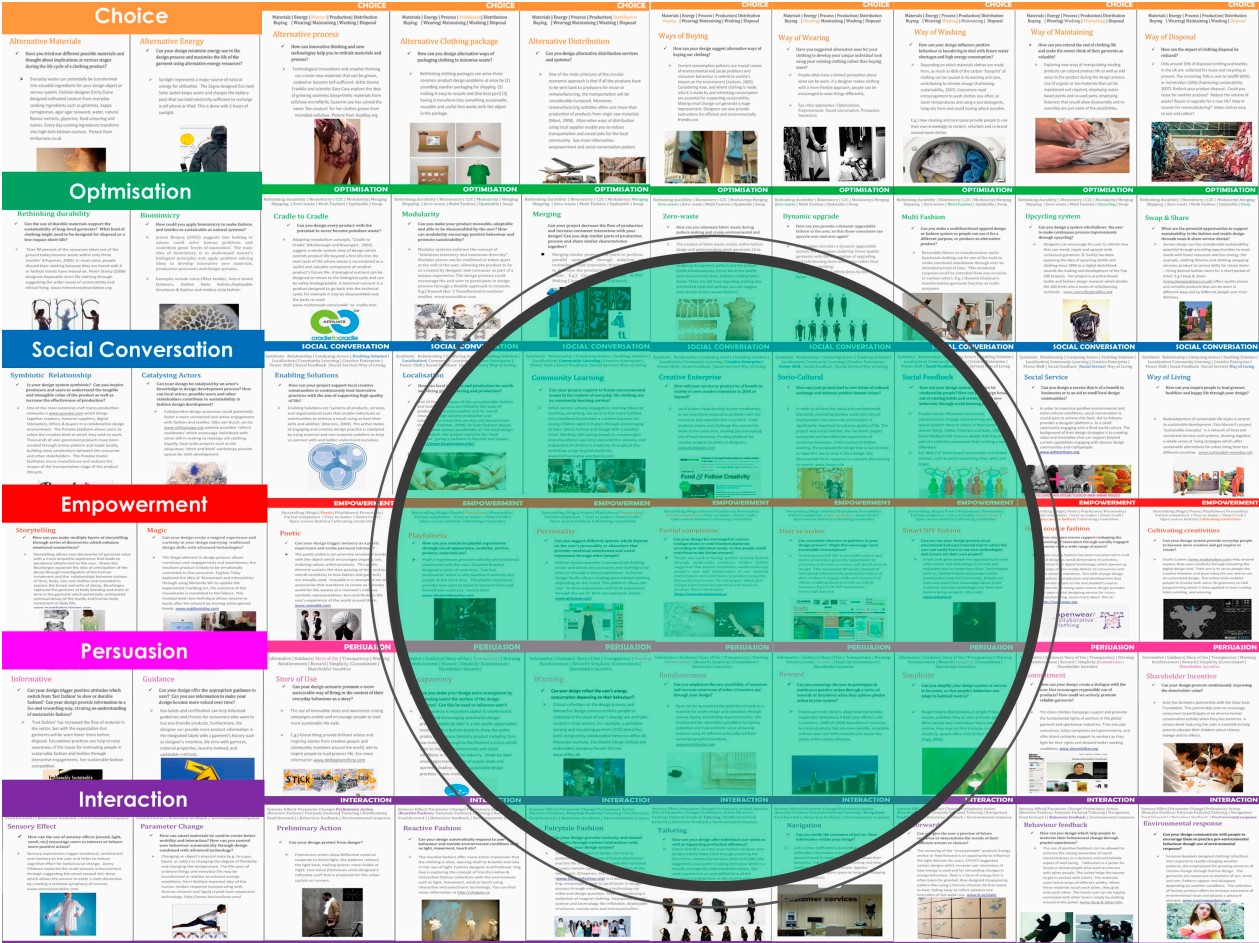

**Figure 2.** SFB ideation cards.

The cards adopt a 'design pattern' [57,59] format. Each card features a pattern title, a description of the pattern as an open-ended question, a visual representation of a possible application of the pattern, and a corresponding description of use. The open-ended questions promote a solution-focused mindset and provide a shared problem space for collective creativity. Examples of open-ended questions from the SFB patterns can be seen in Appendices A and B.

### 2.4.2. Toolkit Content

Each design category comprises a different aspect of sustainable fashion.

The Choice pattern encompasses the entire product life cycle, from production to consumption, and includes the influence of fashion professionals' choices on clothing production and the customer experience throughout the pre-purchase, purchase, and post-purchase processes. While research in these areas is still in its early stages, previous studies have identified structural obstacles such as a limited awareness of sustainable

design strategies [60], a lack of second-hand reuse, repair, and upcycle methods, and insufficient recycling options [61,62]. Optimisation cards aim to reimagine the fashion industry ecosystem and promote a circular economy by proposing flexible production and consumption systems, as well as exploring the application of circular fashion services and system designs. These patterns incorporate concepts such as "cradle-to-cradle" [63], which advocate for extended product lifetimes, and draw inspiration from nature [64]. Social conversation cards combine the principles of social learning efficacy, such as the concept of creative communities, with open-source concepts that inspire individuals to participate locally and globally [65]. Hoolohan and Browne highlighted the importance of design thinking and systemic social solutions that allow practitioners to participate in activities that reflect the nuances of practice-based research and practice-oriented design solutions. Social conversation aims to discover solutions that allow people or communities to work together to solve social problems in a symbiotic way [66].

Empowerment cards assist in the development of products and services that meet people's psychological and social needs by fostering meaningful relationships with clothing users and encouraging them to reconsider their behaviours by providing alternative design options, experiences, and empathy [67]. Persuasion cards use interactive activities to increase the awareness of challenges in sustainable fashion and textiles, and several academics have advocated physical and cognitive strategies, such as the appropriate use of contextual guidance, information, and systems, as design approaches to promote sustainable behaviour [68–70]. Interaction cards seek to minimise the cognitive effort required to function successfully and to provide an automated response by-product or service design, enabling 'intelligent control' through communication between goods and people [70,71]. Our habits and routines shape our behaviour, and consumer behaviour is often the consequence of instinctual, reflexive responses to stimuli rather than deliberate cognitive thought [68,72]. Fashion product services and systems or appropriate communication design could, therefore, influence consumer habits and reduce the intention–behaviour gap in sustainable consumption.

## 3. Research Methods

### 3.1. Design Thinking and Co-Design Workshop

This study adapted the design thinking (DT) models from IDEO's 3I model ('Inspiration, Ideation, and Implementation') [73] and the 4 D model of a double diamond framework [74] when utilising the SFB toolkit. While many researchers have evaluated DT processes and characteristics, few studies have examined how DT can be used effectively to develop sustainable fashion solutions among different types of co-designers. IDEO's design thinking model makes use of three spaces: 'Inspiration' involves identifying a problem or opportunity and linking it with potential solutions. The 'Ideation' space is where ideas are generated and continue to develop through the synthesis of insights. The final stage, 'Implementation', transitions the initial project into the real context of people's lives by visualising the product or service prototypes. Similarly, Britain's Design Council has introduced a 4 D model with a double diamond framework that incorporates divergent and convergent stages of the design process. The framework consists of: (1) discovering and exploring insights into the problem, (2) defining and identifying gaps in the area to concentrate on the issue, (3) developing and creating potential solutions to tackle the identified issues, and (4) delivering a solution that works in the real world [74].

Both DT models are widely adopted by practitioners and design academics. However, there is limited guidance on how designers and co-designers should work together in a co-design workshop to address sustainable fashion production and consumption. Additionally, there is limited research on how different types of participants can effectively contribute and leverage their skills and knowledge for maximum outputs. Without any toolkit and workshop prompting, it could be that co-design participants would simply concentrate on whatever information group members have in common, instead of sharing their unique skills or knowledge of a particular field [19]. Idea blocking can easily occur

when participants cannot express themselves simultaneously during the group ideation process. Participants may also be held back by a 'fear of evaluation' by other participants or moderators, which can have a negative influence on their participation. The SFB toolkit can assist in dispelling idea blocking and facilitate creative thinking and creativity by minimising fears of peer judgments during the group ideation process.

The existing DT models were adapted and applied to four major activities conducted for the co-design workshops. In Step 1, the discovery phase, participants were introduced to the workshop's rules and procedures. A brief explanation of the purpose of the study and the general workshop process was given before the sessions started. Each participant introduced themselves to their group members and shared their own understanding and perceptions of sustainable fashion as well as of the challenges associated with promoting sustainability at the personal level and from a consumer perspective. Each participant then explored the challenges involved in facilitating sustainable fashion production and consumption in the personal, social, and environmental contexts by reading through the SFB toolkit for approximately 20 min. In Step 2, mind mapping and rich picture techniques were used to illustrate individual problem scenarios posing challenges and barriers to practising sustainable production and consumption. With the aid of the SFB toolkit, participants employed visualisation of their ideas to both articulate the problem situation and develop and depict their hoped-for solutions to it.

Step 3 included integrating DT with other design specifications to help develop a sustainable strategy. In this synthesising process, the initial design concept was tailored to the specific target market (e.g., by age, gender, and lifestyle) and built into a concrete concept. In step 4, each team presented their design concepts and discussed their ideas with the other groups to evaluate whether their ideas could be effective ways to develop sustainable fashion product–service systems.

*3.2. Data Collection and Analysis*

Mixed-methods data collection was employed to test the SFB toolkit, consisting of co-creation workshops, observations, questionnaires, and face-to-face interviews. Data were collected during each workshop activity by way of audio recordings, photographs, and field notes taken by the facilitator to capture various aspects of the task. Each workshop lasted two to three hours, during which the ideation activities were conducted, and subsequently the toolkit content and workshop process were evaluated. The initial pilot study was conducted in the UK. The pilot study's objective was to evaluate the workshop process's efficacy, estimate its duration, and make additional modifications. After this pilot study, several co-design workshop activities were undertaken with fashion design students ($n = 35$), mixed multidisciplinary design students ($n = 17$), fashion enterprise master's degree students ($n = 28$), and fashion marketing students (52); all workshops used the above-mentioned 3I and 4 D DT techniques for co-design. The final study was carried out at the British Council in Seoul, Republic of Korea, with one design manager, two fashion entrepreneurs, two fashion designers, and five members of the general public. Table 1 presents an overview of the workshop participants' information and includes a brief overview of the data collection methods.

**Table 1.** Overview of Workshop Participants' Information and Data Collection Methods.

| Workshop No. | Participant Information |
| --- | --- |
| 1 | A co-design workshop with 5 participants of mixed backgrounds |
| 2 | A workshop with 17 fashion design (FD) students in the UK |
| 3 | A workshop with 18 FD students in the UK |
| 4 | A workshop with 17 design master's students in the UK |
| 5 | A workshop with 28 fashion enterprise (FE) master's students in the UK |
| 6 | A workshop with 27 fashion marketing (FM) students in the UK |
| 7 | A workshop with 25 FM students in the UK |
| 8 | A workshop with 10 fashion-related business start-up entrepreneurs and students in Seoul, Republic of Korea |

A total of eight workshops were conducted, and 147 people participated. All participants were given a paper-based workshop guideline, coloured pens, and A1-sized paper. Each participant also received an A4 piece of paper for the individual brainstorming activity. Participants were grouped into teams of five to seven people, in which each played the role of one of the members of a creative team, to simulate the real-world design environment. Adapting Bell and Morse's image content analysis, the participants' rich pictures and generated concepts were analysed, along with their interview discussions. Audio-recorded interviews were transcribed into text [75]. Braun and Clarke's thematic analyses were used to find recurring patterns and meanings in a qualitative research context. The six phases of analysis include: (1) familiarisation of data, including interview data and user-generated concepts and observation field notes; (2) the generation of initial codes; (3) the search for themes; (4) revision of the themes; (5) defining themes' names; and (6) reporting research findings, discussing the results, and comparing them with the existing literature [76].

### 3.3. Interview-Based Evaluation

Semi-structured face-to-face interviews were conducted with various fashion professionals including one head of CSR and sustainability, one creative director for a fashion and home brand, three marketers, a fashion business consultant, one fashion designer, and five design educators. Table 2 summarises the backgrounds of the participants.

**Table 2.** Interviewees' backgrounds.

| Interviewees | Professional Background |
| --- | --- |
| Person (P) 1 | Head of CSR and sustainability from the outdoor sector |
| P2 | Creative director from a fashion textiles, homes, and jewellery brand |
| P3 | Senior marketer (senior account executive) from a fashion public relations and branding agency |
| P4 | Marketer (social media executive) from a public relations firm |
| P5 | Marketer from a luxury fashion brand |
| P6 | Fashion designer from a business-to-business consultancy |
| P7 | Fashion design lecturer from a university |
| P8 | Fashion marketing lecturer with several years' industry experience of running fashion companies |
| P9 | Sustainable design lecturer |
| P10 | Fashion designer from a men's clothing accessories company |
| P11 | Fashion management lecturer with several years' experience as a fashion buyer |
| P12 | Design management lecturer |

The interview subjects were provided with the SFB toolkit, and the interviewer described its purpose and application in the workshop. In-depth semi-structured interviews were then carried out to solicit participants' perceptions of the quality of information, usability, layout, advantages, and limitations, and suggested improvements. Each interview lasted approximately 30 min to 1 h and was audio-recorded with the participants' consent. The interviews were transcribed in advance. Thematic analysis (Braun and Clarke, 2015) was employed for analysing the interview data.

## 4. Results

### 4.1. Evaluation of the Toolkit and Workshop Process

The participants reported that the toolkit helped to generate concepts of sustainable fashion. The tool cards acted as inspirational MCSs for discovering alternative sustainable fashion solutions for PSS design. Each tool card question assisted in building a foundation of knowledge on sustainability and for learning different ways to approach the sustainability project. A total of 85% of participants responded that they would use the toolkit again, and 85% said their knowledge and awareness had increased. This contrasts with 11% who said they would not use it again and 4% who were unsure whether their knowledge had increased due to the workshop. Among the negative comments, the participants said it had been difficult to pinpoint the issue of sustainability in fashion and create new ideas due

to time constraints. An additional clear directional tool was vital for toolkit users, since it would allow them to make use of the instructions without a facilitator's support. The Choice pattern (26%) was rated as the most beneficial, followed by the Optimisation pattern (21%) and the Social Conversation pattern (20%). The Interaction (18%), Persuasion (12%), and Empowerment (8%) patterns received lower ratings than the other tool cards. The open-ended responses, on the other hand, demonstrated that the majority of participants thought all of the contents of the ideation toolkit were of equal value and that the majority of its sections were helpful. The participants reported that by mixing several cards, it was simpler to pinpoint issues, consider their design strategy in greater detail, and swiftly map out problem-solving pathways. Most participants responded that they utilised the toolkit to identify problems through the open-ended questions and to generate alternative solutions through the reading of other instances of the same design problem on the various cards. One of the FD participants noted, "*It helped me to consider the points that haven't occurred to me before (for example, I used to concentrate on the materials choice only, but now see it just like a small part of the whole sustainability concept). I think that it also helped me to link ideas that at first might have seemed totally unrelated*". It was acknowledged by some participants that when they first reviewed the information in the toolkit, they believed some cards would not be relevant to sustainability, particularly the Persuasion and Empowerment pattern cards. However, as time went on, they discovered these patterns were highly beneficial in helping to generate fresh ideas for sustainable PSS design. Appendix C provides a summary of participants' perspectives on each card pattern.

Participants visualised the key sustainability challenges in relation to current social, cultural, and environmental issues. Figure 3 showcases the SFB ideation cards and examples of participants' mind mapping activities.

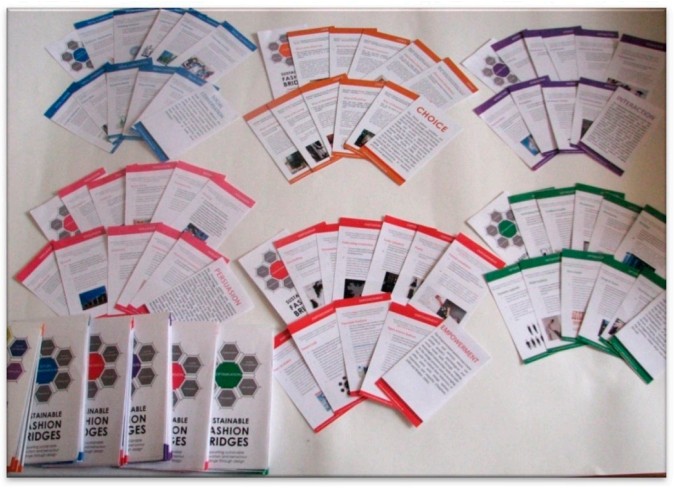
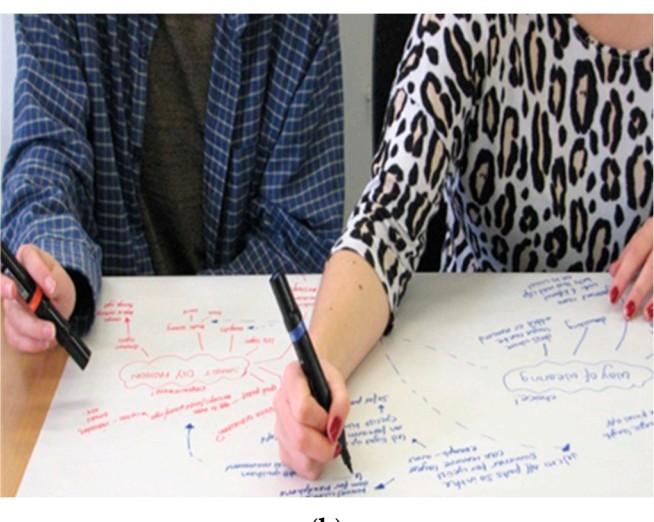

(**a**)            (**b**)

**Figure 3.** The SFB toolkit task analysis: (**a**) Exploration of the toolkit; (**b**) Idea mapping process in the co-creation workshop.

When accessing the toolkit, participants also utilised general mind mapping and rich picture visualisation techniques to help communicate their thinking and ideas to group members. The most-cited advantages of using a mind mapping activity were that it was easy and helped participants to synthesise their ideas and see complex relationships. One of the FD participants noted:

> "*The mind mapping process helps narrow down the specific problems through writing down the keyword. It allows the identification of the key solution to work on a new concept logically. It really helps to bring out lots of ideas from the ideation cards and write down a core solution through the mapping system. It identifies key elements and*

*looks at the co-relation of each idea, which triggers innovation and helps us to bounce ideas off each other"* (W2).

The main advantage of using rich pictures was to encourage the flow of ideas and stimulate discussions within the group. It aids in defining key problems by making visual associations between different actors and components in the PSS. One participant from the FM students commented, *"Rich picture building encourages creativity and the development of more ideas compared to a simple mind map"*.

However, the participants also mentioned some disadvantages of using rich pictures. One participant from the FE students said, *"The rich picture tool may take longer to get used to than mind mapping. It can also be challenging to think outside the box once the rich picture is created"*.

Some participants found it initially challenging to create solutions for the long-term future as this was a less familiar procedure. They struggled with predicting the future, but the activity had the advantage of encouraging them to think forward. The overall feedback on the visualisation process was positive, although some participants mentioned being challenged in formulating their perspectives and expressing their thoughts on paper.

One of the FM participants (W6) responded, *"It was a bit bizarre to begin with, as the ideas that we brainstormed seemed so unrealistic; however, as time went on, it was easier to do"*. Participants responded that the final group discussions were very useful, as they helped them identify alternative perspectives. Figure 4 shows examples of discussions of each group's concepts during the final idea review stage.

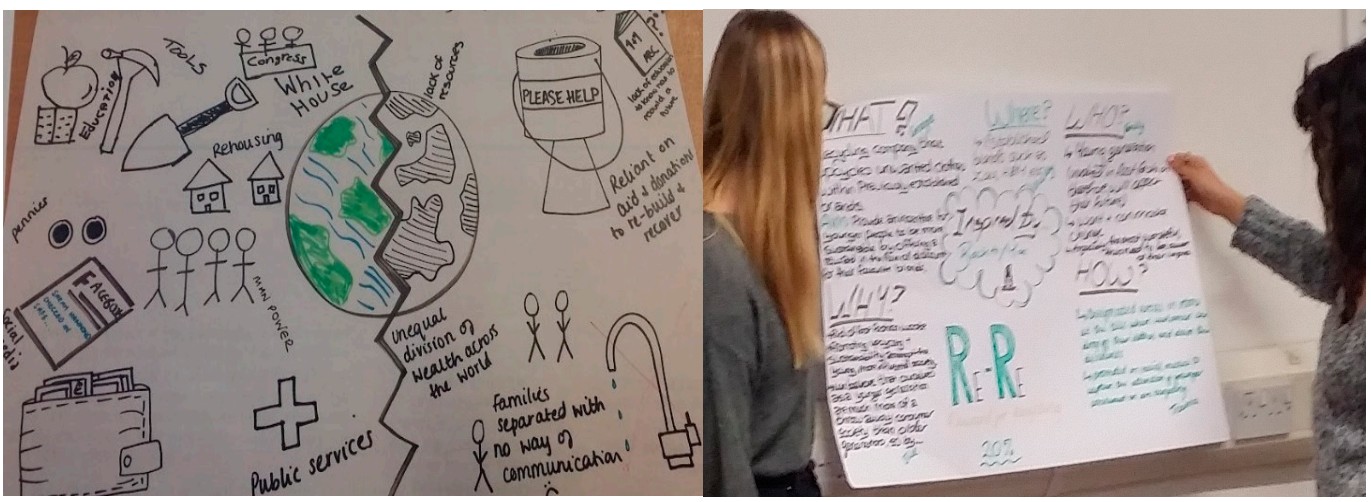

(**a**) Rich picture building            (**b**) Group discussion

**Figure 4.** Rich picture building and group discussion.

Most participants reported that the co-design method strengthened the development of design ideas and that it helped them re-evaluate sustainable fashion across multiple spectrums by gaining insight into others' perspectives. The co-design approach was deemed helpful for developing an effective design concept by 84% of participants; 10% said it had "no benefit" and 6% said they were not sure. The comments revealed that participants found the process to be instructive and inspirational. They emphasised the positive outcomes of group learning and knowledge sharing. By contrast, participants also found the tasks challenging at times, especially when members of the group had diverging opinions. An additional challenge of the co-design process is that it can be hard to get people equally involved in the ideation process. Participants had varying levels of knowledge about sustainability in fashion, and for those who were new to the topic, additional time was needed to grasp the sustainability-related issues in fashion design.

### 4.2. Interview-Based Evaluation

In accordance with the responses to the workshops, interviewees identified that the SFB toolkit's greatest asset was its capacity to explore sustainable fashion in a more creative manner by combining two or three cards to create fresh, compelling ideas. The respondents' opinions on the toolkit's potential practical utility for fashion enterprises or design consultancies, such as for training purposes or the creation of fresh strategies based on sustainability, were largely positive. Interviewee P11 stated,

*"A lot of companies and organisations need to take knowledge from the design sphere. For the design consultancy, a more multi-disciplinary team including designers, pattern makers, technologists, buyers, merchandisers, marketers, and managers sitting down and developing strong design strategies would be very valuable. They often need to develop a better understanding of where they are going and what the real future concerns are". (P11).*

Interviewees from industry felt that the toolkit could be useful both for fashion students and for industry practitioners. One marketing executive said:

*"I think it is thought-provoking, especially if you are in a brand or if a lot of brands need to reposition the way their internal business strategy is, for it to be sustainable. If they chose to start it small, they wouldn't need to go through every single option. Even when they look at one set of cards, they can positively impact future environments. There is so much pressure on brands everywhere to be more sustainable, to change the way that their business works. A lot of people have no idea where to start, but this makes it really simple, so you can just pick a card or flip through trying to find ways to adjust your business to be more sustainable"* (P4).

The SFB Ideation toolkit includes elements of multi-disciplinary strategies to design a sustainable PSS design through promoting critical thinking, system thinking, future forward-thinking, innovation, and collaboration.

Interviewees highlighted four major benefits of the toolkit: (1) enabling critical thinking by offering problem-solving mechanisms pertaining to environmental and social issues throughout the clothing life cycle; (2) facilitating an understanding of the impacts of holistic system–thinking and how each action can influence the rest of the supply chain; (3) promoting conversations about sustainability challenges in fashion PSS design and inspiring ideas for potential or alternative future practices; and (4) aiding a group of individuals in developing communication and consensus during the co-design workshop process. Some sample responses are shown in Table 3.

### 4.3. Suggested Improvements and Other Aspirations for Tool Usage

Although interviewees considered that the toolkit offers potentially positive benefits, as described above, there are several points at which there is room for improvement in the SFB tool. The key development needs include: (1) providing a clear purpose and outlining the benefits of addressing sustainability in the fashion industry, (2) effective instructions and clear directional guidance on toolkit usage without a facilitator's support, (3) triggering actions for future sustainable design practices tailored to various user situations, (4) visual content that inspires users to stimulate their imagination and further discussion, (5) the development of an easily accessible digital toolkit, and (6) the integration of critical reflection and evaluation steps to arrive at the best solution and to know whether user-generated concepts generated using the toolkit are offering sustainable values. Table 4 provides a summary of additional suggestions from industry professionals and educators for the future development of the toolkit.

**Table 3.** Roles of sustainable fashion toolkit for the ideation.

| Themes | Sample Responses |
|---|---|
| Critical thinking: enabling development of critical thinking by offering problem-solving mechanisms regarding environmental and social issues in the overall clothing lifecycle | • *'I really like questions on the reflection that make you think, and it is a good place to start. I also like different types of options because we are working with various companies and different brands fit into a lot of different categories. I think having more options gives you a more in-depth analysis. Especially with sustainability, there is not one way alone to be sustainable'* (P3).<br>• *'I think the toolkit is very useful for fashion designers and students particularly to make them aware and explore the possibility and marriage of different aspects of sustainability that they can be achieved. In terms of industry, again they can identify how they can enhance sustainability within the company and also identify limitations. Each time you look at sustainability it will be likely more incorporated sustainability'* (P7). |
| System thinking: helping understand the impacts of holistic system–thinking and how each action can influence the rest of the supply chain | • *'I can see the context of an educational tool for designers in making them aware of systems thinking approach'* (P11).<br>• *'A lot of books suggest fashion design products rather than think how they can be made different as a holistic aspect. A company may not even think about any other possibility, for example, they might have thought about environmentally friendly materials use but not many other options are considered. I think it can create awareness of all the different aspects and support to create a new capability for overall design'* (P7). |
| Creativity and innovation: facilitating discussion of sustainability issues in fashion PSS design and triggering thought of what potential or alternative practices could exist for the future | • *'I would be totally interested in the tool. As a company, we are always trying to find ways to be sustainable. It could be a very powerful tool nationally and internationally for schools, lectures, and SMEs'* (P2).<br>• *'I think the combination of the toolkit with future scenario building is a very strong point of this workshop process. I think it is not only for use at the idea generation stage, once used in the professional situation, but it can also be a good strategy for companies. If you got people, sitting around the table, who are normally arguing precise quality parameters, it would have the potential to take out that point to let them think that we are in the 2030s and to start sharing pictures where their business might go, I think that is very useful point'* (P11). |
| Collaboration skills: supporting the creation of dialogue and communication of shared understanding within a group of people during the co-design workshop. | • *'I think there are a lot of organisations and designers that can actually build in time for incorporating sustainability into the design process in the same way you tested. Although it seems more useful to a group of people rather than an individual, the toolkit could be assessed during the individual concept generation stage'* (P8).<br>• *'The information on the cards sets is very easy to understand .... I can see how this inspires discussion among the people to develop ideas. I think the layout of information is good'* (P8). |

**Table 4.** Suggested improvements and other aspirations.

| Themes | Sample Responses |
|---|---|
| Providing clear benefits: outlining the values of addressing sustainability in the fashion industry | • *'The toolkit offers several alternative scenarios and examples, but industry knowledge has increased massively on sustainability. Part of the tools needs to be about explaining why we need to do this. There are a lot of policies at the moment about circular economy and sustainable product development, so maybe it is worth linking into the drivers'* (P1).<br>• *'If a designer can see the benefit or value of the toolkit use, such as why they should use it and why this kind of process would be more beneficial to them, the professional designers will make time to use the toolkit as it supports the development of designers' ideas integrated with sustainable fashion'* (P8). |
| Effective instructions: clear directional guidance on toolkit usage without a facilitator's support | • *'The current situation of the fashion design business model is driven by increasing economic value. It will be very challenging if the process is too complicated and if a lot of effort is needed to incorporate this'* (P12).<br>• *'It would be better to make a clear description of the idea generation process with the toolkit so that users can follow the task and look at this inside the instructions. The potential users could use the toolkit without a facilitator, and they can also use the toolkit during the individual idea generation process'* (P9).<br>• *'A possible card sorting process whereby the user can use two sets of cards or three set of cards, or just let them select some specific part to allow them to explore it. If they are given some guidance on how they would select those cards in the first place, they would easily utilise the cards'* (P8). |

**Table 4.** *Cont.*

| Themes | Sample Responses |
|---|---|
| Personalisation and interactive tool: triggering actions for future sustainable design practices tailored to various user situations | • *'Though the tool card briefly summarises each section, it will be difficult to understand the whole concept of sustainable fashion through a one-day workshop. It would be much better if the tool cards have some more illustration regarding terminology and overall meaning of sustainable fashion'* (P9).<br>• *'If we can search words of the documents, interactive search terms to find relevant information quickly and easily. It will show you every time you have mentioned it in the document'* (P3). |
| Stimulating inspirations: visual content that inspires users to stimulate their imagination and encourage further discussion | • *'I think the toolkit needs to be more visual, especially if the toolkit is targeted at designers'* (P9).<br>• *'Your brain is sometimes better at reacting to images than words. Flow charts, visualising ideas through drawing, are an important part of ideation'* (P3).<br>• *'When people are provided with a lot of choices, it becomes overwhelming. People often have screen fatigue and information overload. As creatives, we tend to gravitate toward visual communications opposed to written communications. Quizzes or visual scenarios would create a bigger impact and engage it more. The text should not be overly wordy or academic'* (P2).<br>• *'Double-sided card type in which the front side provides the open-ended questions while the reverse side shows a short description of the image and description... Perhaps, when users start to build ideas through the visualization process, other pieces of cards can be provided that are white and clean. Users can put their own image on the reverse side, and they can use these various images for future use to help as a creative process for themselves'* (P8). |
| Easily accessible digital tool: (e.g., a digital app or an interactive web platform) that could be more effectively employed to help individuals or teams to generate ideas. | • *'I think definitely a digital app that makes it more interactive. You can also see ever-growing updating and add more case studies, ideas, and examples of how people utilise the tool within the industry in different scenarios you have and categories'* (P5).<br>• *'I think online, flash digital cards, digital can reach wider audiences'* (P1).<br>• *'An online platform can be very useful, but it will be more beneficial if the process and platform can be like a sophisticated game or enjoyable environment where they can enjoy practices rather than work. Sometimes sustainability is considered as very hard work or an enormous task'* (P8).<br>• *'If you create an online platform that allows stockholders to work on a different section of the supply chains and inputs and outputs, there could be different check points where they could make that strategy better'* (P4). |
| Offering a reflective tool that enables an evaluation of toolkit users' end outcomes for sustainable fashion practices | • *'Creativity can get complex losing focus of the initial objective. The outcome of the design concept can be something completely different... Especially, design students are outcome-oriented; people often jump to solutions before really understanding the problems. Maybe a reflective tool that users can evaluate their end outcome that captures where the sustainable impacts are and what the change is and where the change is'* (P11). |

## 5. Discussion

### 5.1. Roles of MCSs in Sustainable Fashion Innovation

A new sustainable fashion innovation tool can alter user perception of what defines value propositions and introduce users to a larger range of sustainable fashion value creation opportunities. Yang observed that several companies have limited perspectives on value creation, concentrating mainly on customer value (what is valuable from the viewpoints of their consumers) and economic value for the business [77]. Conventional norms of sustainable fashion often use a product-centric approach, and other types of innovations [78], such as developing new structures, product systems, processes, performances, channels, and consumer engagement and profit models, have not been investigated for their usefulness in diversifying the range of sustainable fashion innovation. One important skillset for sustainability entrepreneurs is dealing with the multifaceted, systemic connections of innovation systems [79]. The SFB tool could facilitate multiple ranges of sustainable fashion innovations in the ideation stage for future sustainable fashion professionals.

This study reveals that the effectiveness of the toolkit is critically related to its instructions and usage processes. As Davila and Ditillo have recognised the importance of the MCS in managing the collective creative process, this study provides an application of the SFB tool to show how this inspirational MCS can be used alongside a more directional MCS in the design innovation process [29]. The lessons from the workshops and interview results indicate that the toolkit serves as an inspirational MCS, but there is limited support

for effectively integrating the tool with directional MCSs and for users to critically evaluate their concepts. Directional MCSs help to define overall project boundaries and synthesise the key concepts after embedding into them critical reflective elements to find the best solution to address sustainability, whereas card-based tools help to minimise roadblocks during the ideation process as a form of inspirational MCS.

Based on feedback from workshop participants and interviewees, additional toolkit instructions and key steps in the idea generation process were developed to support both inspirational and directional MCSs (Figure 5). The SFB directional MCS can serve as directional MCSs that facilitate convergent thinking by establishing common sustainability objectives, scopes, and directions. Its directional instructions help manage the overall ideation process to facilitate convergent thinking and critical reflection. The final stage of the ideation process is group reflection upon and evaluation of the ideas, during which the design team can score the concepts before the final execution of the design (see Appendix D for the SFB evaluation checklist). After the final idea is selected, the project team can set a clear project direction and proceed to implement their concept by developing the prototype, experimenting, and conducting user testing.

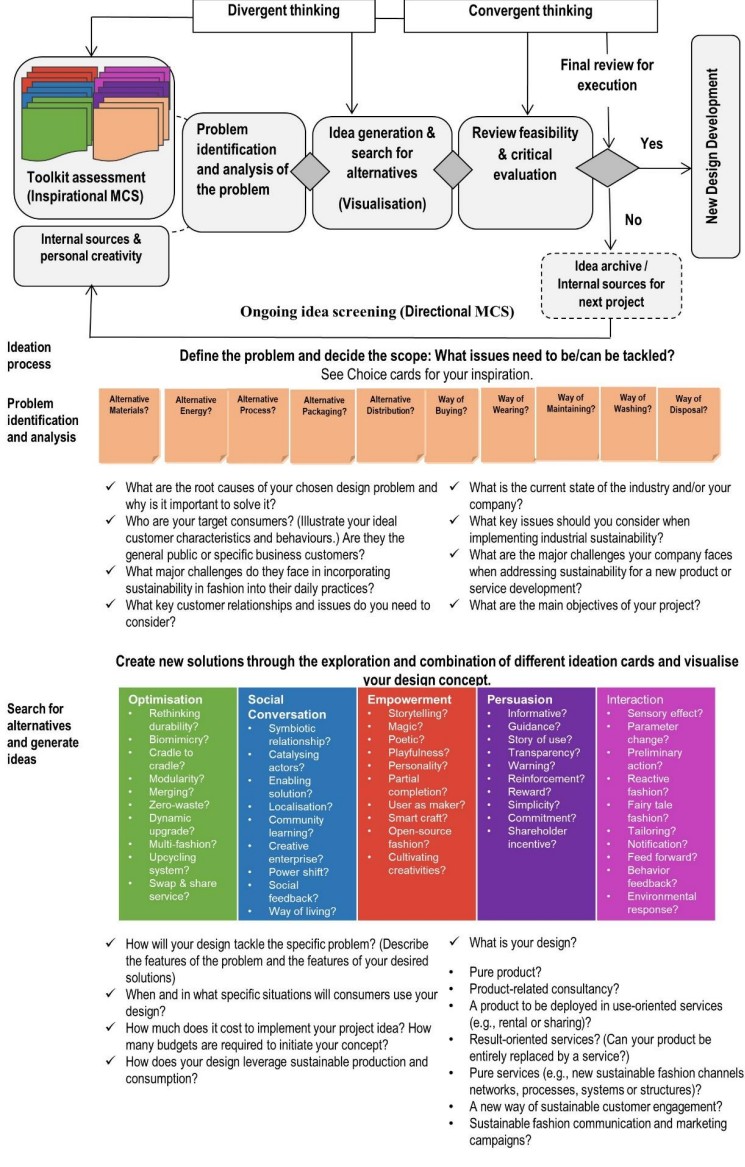

**Figure 5.** SFB directional MCS.

### 5.2. Sustainability Skills and Sustainable Fashion Education

Over the last few years, the job market for students from fashion-related courses including design, marketing, and management has shifted dramatically. As fashion brands and retailers struggle to meet increased sustainability standards and expectations, the introduction of a new raft of educational and training measures is vital to building the sustainability knowledge and analytical and critical skills necessary to develop innovative solutions [61]. Various commercial and educational organisations in the fashion sector demand the incorporation of sustainable fashion design practices into the curriculum. Nonetheless, there is very little in the way of empirical case studies and practical investigations into how designers and other sustainability stakeholders can effectively utilise sustainable design tools and what key barriers are commonly encountered during the idea generation process.

In this rapidly changing market environment, several fashion companies and higher education sectors are trying to reform sustainable fashion education and training to build capabilities that can meet the expectations and increased pressure of sustainability standards and the Industry 4.0 revolution. González-Pérez and Ramírez-Montoya stressed the importance of more opportunities for lifelong learning for all as inclusive and equitable education that will mitigate the harmful effects of the post-pandemic era's environmental and social issues [80]. We examined how the SFB tool can help develop the skills and knowledge necessary for a new generation of interdisciplinary sustainable designers and strategists. This study offers an application of the SFB tool that promotes Hargadon and Bechky's positive social interactions, raising the profile of design-based approaches among other sustainability stakeholders [34]. The toolkit assists in the informal and structured management of the quest for alternative solutions, of ideas sharing, and of the critical reflection of our design practices. The key contribution of this research is its provision of guidelines for the optimal development of a sustainable design tool for addressing both production and consumption in the ideation stage.

Rana and Ha-Brookshire recognised three core skills for ideation in fashion design: creativity, critical thinking, and collaboration [36]. Our study's findings show that these three skills are highly valued, but that fashion professionals also highlighted the importance of the sustainable fashion innovation of systems and strategic thinking skills that enable the understanding of how product design can be interlinked with the wider supply chain system, society, and culture. The participants also emphasised that future-forecasting and problem-solving skills are needed to prepare for uncertainty and to be more resilient.

### 5.3. Using Design Thinking and Co-Creation for Sustainable Innovation

The Literature Review section highlights the limitations of card-based tools, including that they are often static, making it necessary to consider how to minimise tension between the amount of information a tool provides and the evolving imperatives of creativity. This study's findings show that toolkit data can be employed based on users' levels of sustainability and design implementation skills to leverage maximum outputs in sustainable fashion innovation. In other words, co-creation users of various backgrounds can use the SFB tool based on their levels of sustainability skills and design implementation experience to elicit maximum results. Beginners who lack knowledge of sustainability tend to prefer highly visual information that is easier to understandable, while advanced users want more-detailed data, which the case studies in the cards provide. Figure 6 shows how various types of users in a co-creation workshop can use the SFB tool. Four major types of co-designers were identified, as follows:

(1) Sustainability beginners with low levels of sustainability skills and no design experience or design implication skills.
(2) Sustainability analysts with high levels of knowledge and skills in sustainable fashion, but no design experience or design implication skills.
(3) Fashion products–service systems (PSS) designers with low levels of knowledge and sustainability skills, but high levels of experience in design implementation.

(4)    Sustainable fashion innovation specialists with high levels of sustainability skills, and engagement in sustainable fashion and design implementation.

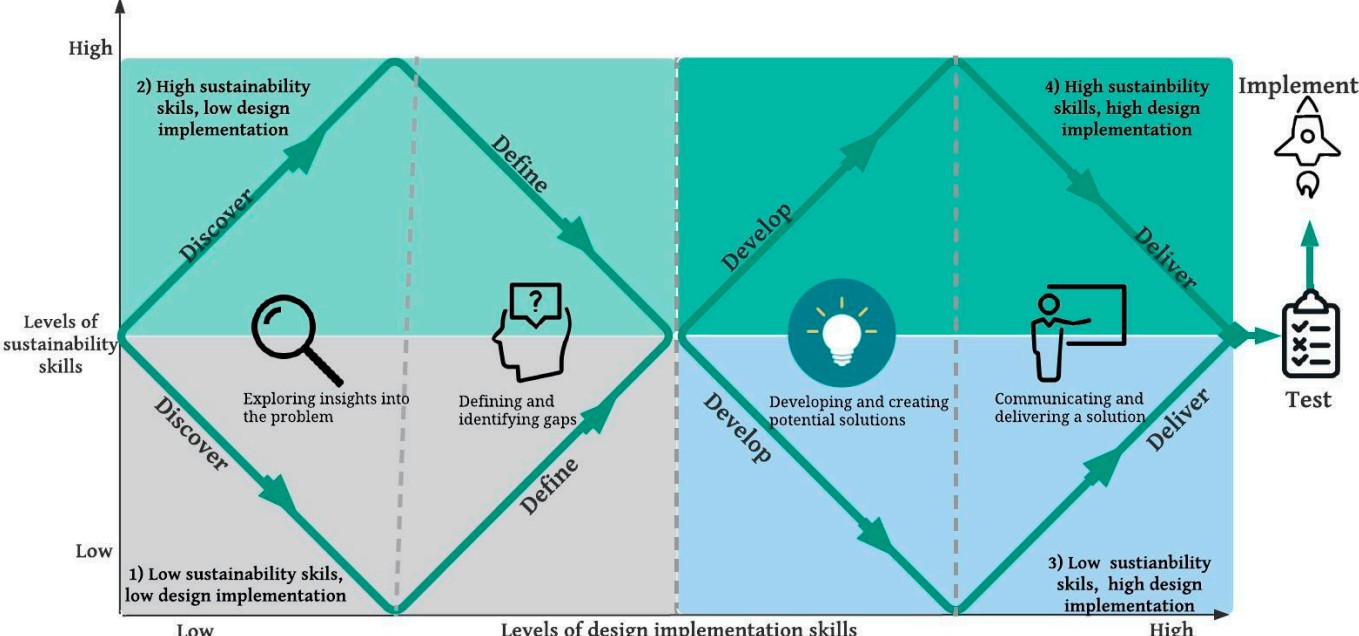

**Figure 6.** Sustainable Fashion Innovation Matrix and roles of MCSs in showing the relationships among levels of expertise in design implementation and sustainability skills.

(1)    The beginner level for co-creators: Those who have little or no experience participating in design implementation activities and who have a low level of knowledge and skills in sustainable fashion. This group can share their experiences with barriers to pro-environmental actions and their perceptions and experiences of sustainable fashion products and services from a consumer perspective. General consumer groups can offer valuable insights during the discovery stage of the DT process by sharing their product and service experiences, issues, and their needs for sustainable fashion. A number of possible issues could be explored using divergent thinking at the beginning of the design process by looking at the issues related to sustainability in fashion production and consumption and consumers' behaviours with the fashion products and service interactions.

(2)    High sustainability skills and low design implementation: In this group, participants may have a good ability to engage in pro-environmental behaviours and have good knowledge of sustainable fashion, but not much experience with design practices. Participants can define the key issues of sustainability as they relate to fashion. Convergent thinking can be used to narrow down and define key problems by identifying themes and finding connections and relationships with the issue. In this stage, co-design participants can explore design implementation possibilities to tackle the identified problem.

(3)    Low sustainability skills and high design implementation: People in this group generally have low levels of knowledge of and engagement with sustainable fashion, but high levels of experience in design implementation. Some fashion design participants expressed that they would have preferred to move directly on to the next step in the fashion design process, with a hands-on experience of garments to assist in the generation of solutions. The making and crafting process is a pivotal aspect of fashion design implementation. Reflecting the consequences and environmental and social issues of the fashion production process, co-designers could implement better design solutions by addressing environmental and social impacts throughout the clothing life cycle.

(4)  High sustainability skills and high design implementors: Participants described in the top right quadrant are fully conversant in both sustainability issues. They also have the ability to implement the conceptual sustainability ideas of design practices in real-world contexts. Designers can share their knowledge and skills with users by actively involving rapid prototyping and micro-production services.

*5.4. Future Directions for the Sustainable Fashion Innovation Tool*

The development of sustainable fashion services, experiences, and systems design is still in its early stages. Future co-design practices could involve active engagement with textile and clothing manufacturers to implement product and service design. These tools would be particularly beneficial for general audiences and newly graduated fashion students, encouraging and motivating them to practise sustainable fashion. Further research could explore how user-generated concepts formulated with the toolkit can be implemented and marketed in the fashion industry. Beyond its application in fashion and textile design education as a teaching resource and learning tool, the developed toolkit could serve as a practical catalyst for the industry to address sustainable fashion challenges by discovering creative approaches to tackling sustainable fashion PSS challenges within a co-design process. Additionally, the roles of aesthetic and craftsmanship elements in co-design could be investigated to understand how sustainability can balance functional and aesthetic aspects in the process.

The empirical findings of this study were obtained in an educational context, primarily involving design, marketing, and fashion entrepreneur students, as well as a small group of professional designers and fashion entrepreneurs. The workshops lasted approximately two to three hours, and the evaluation process was conducted in a condensed format. It can be argued that this process may not fully represent how the toolkit would be used in the fashion industry. Therefore, studies could be conducted to explore variations in toolkit usage among different types of fashion companies. Longitudinal studies could also consider the unique situations of these companies to identify potential gaps in their ability to implement sustainable design.

Another significant challenge in utilising existing card-based tools is that card tools can oversimplify the facts, and the recommended information in the tool may become outdated as the field evolves [56]. Any paper-based tool information can be dated as industry and consumer needs are continually changing and additional environmental issues continue to be raised. Therefore, future research could be carried out on the development of an interactive co-design online platform that could help users contribute to knowledge generation and share their own challenges of incorporating sustainability in their daily practices. New forms of knowledge could then continuously evolve, co-created by users.

Therefore, future research could be carried out on the development of an interactive co-design online platform (e.g., in a digital app version or on an interactive web platform, as suggested by workshop participants and interviewees) that could help users generate ideas in the design process. The toolkit could be made available to audiences by either an online card-based tool download or a web or mobile interface to access the overall project background along with information such as key terminologies, background research on sustainable fashion, the specific theory behind each pattern, and examples of user-generated concepts, regardless of geographical location. Further, an online co-design platform could potentially be of significant benefit by forming networks of co-designers drawn from wider audiences—from other stakeholder groups, say, or from consumers themselves—thus leveraging sustainable social innovation to generate interactive, user-led designs. Moving beyond the mere online provision of information and the associated opportunities for interactivity, the toolkit could even be digitised at a more sophisticated level by integrating it into a game, where users can interactively select and play SFB cards based on their levels of knowledge and intended goals. Game-based learning has the potential to overcome the overwhelming challenges associated with incorporating sustainability into the fashion

process by positioning the interaction of sustainable fashion goals with creativity and user experiences at its core.

## 6. Conclusions

This study investigated the key sustainability information and criteria to consider when creating a sustainable fashion tool for the ideation process. We examined the roles of co-design and toolkits in promoting and managing sustainable fashion innovation. The study contributes to the body of knowledge about sustainable fashion innovation by: (1) conducting a critical literature review of the concept of sustainable fashion innovation and the levels of sustainable innovation tools, (2) defining the roles of MCSs in providing sustainability information during the ideation process performed by fashion sustainability professionals, (3) conducting several co-design workshop case studies in the context of fashion education, and (4) providing suggestions regarding the future of sustainable fashion innovation tools based on in-depth interviews with industry professionals and educators.

The study's results provide noteworthy insights into the development of tools that foster sustainable fashion innovation and how future sustainable fashion professionals and educators can maximise the benefits of co-creation while developing sustainability skills to further their careers. Closing gaps regarding such insights and skills is a valuable research contribution to the development of both theory and practice in the idea-generation process and the design of sustainable fashion products and services.

**Author Contributions:** Conceptualization, E.H. and K.B.; methodology, E.H.; validation, K.B.; formal analysis, E.H.; investigation, E.H.; data curation, E.H.; writing—original draft, E.H. and K.B.; writing—review & editing, E.H.; visualization, E.H.; supervision, K.B.; funding acquisition, E.H. All authors have read and agreed to the published version of the manuscript.

**Funding:** This work was supported by the Economic and Social Research Council (Grant number: ES/T501955/1) and by a Fashion scholarship from the University of Leeds. The APC was funded by UK Research & Innovation Open Access Block Grant.

**Institutional Review Board Statement:** The study was conducted in accordance with the University of Leeds and was approved by the Institutional Review Board.

**Informed Consent Statement:** Informed consent was obtained from participants involved in the study.

**Data Availability Statement:** Additional data and supporting findings are available in the Appendices A–E. The data is anonymised, and only aggregated information is used to ensure the privacy of the participants. For any additional questions, please feel free to contact the corresponding author.

**Acknowledgments:** The authors greatly appreciate the individuals who participated in this study and offered valuable suggestions for this research.

**Conflicts of Interest:** The authors declare no conflict of interest.

## Appendix A. Open-Ended Questions: Choice, Optimisation, and Social Conversation Patterns

| | Patterns | Open-ended questions |
|---|---|---|
| **Choice** | Alternative Materials | • Have you tried out alternative sustainable materials and thought about implications of your choice for the overall clothing life cycle? |
| | Alternative Energy | • Can your design minimize energy use in the design process and maximise the life of the garment using alternative energy resources? |
| | Alternative Process | • How can innovative thinking and new technologies help you to rethink materials and process? |
| | Alternative Packaging | • Have you tried alternative ways of packaging clothing to minimize wastes and packages? |
| | Alternative Distribution | • Can you design alternative distribution processes, services and systems for your products? |
| | Ways of Buying | • How can your design advocate alternative ways of buying our clothing? |
| | Ways of Wearing | • Can your design support users to appreciate a unique individual look buying used clothing rather than buying more new goods? |
| | Ways of Maintaining | • How can you extend the length of clothing life and make the clothing user think of their garments as more valuable? |
| | Ways of Washing | • How can your design influence positive behavior in laundering to deal with future water shortages and high energy consumption? |
| | Ways of Disposal | • How can the impact of clothing disposal be reduced? |
| **optimisation** | Rethinking Durability | • Can the use of durable materials support the sustainability of long-lived garments? What kind of clothing might need to be designed for disposal or a low-impact short-life? |
| | Biomimicry | • How could you apply biomimicry to make fashion and textiles as sustainable as natural systems? |
| | Cradle to Cradle | • Can you design every product with the potential to never become pointless waste? |
| | Modularity | • Can you make your products adaptable and able to be disassembled by the users? How can modularity encourage positive behavior and promote sustainability? |
| | Merging | • Can your project decrease the flow of production and increase consumer interaction with your design? Can you skip similar parts of the production process that share similar characteristics? |
| | Zero-Waste | • How can you eliminate fabric waste during pattern making and create environmental and economic benefit? |
| | Dynamic Upgrade | • How can you provide a dynamic, upgradable fashion to the user, so that those consumers can upcycle over and over again? |
| | Multifashion | • Can you make a multifunctional apparel design or fashion system so people can use it for a different purpose, or can you produce an alternative product? |
| | Upcycling system | • Can you design a service or system that allows the user to make continuous process improvements through upcycling? |
| | Swap & Share Service | • What are the potential opportunities to support sustainability in the fashion and textile design through swap & share service design? |
| **Social Conversation** | Symbiotic Relationship | • Can your design promote positive symbiotic relationships between manufacturers and consumers? |
| | Catalysing Actors | • Can your design be catalyzed by various people's knowledge in the design development process? |
| | Enabling Solution | • Can your design help creative communities with the aim of supporting sustainable fashion? |
| | Localisation | • Can your design enhance the value of local products and production? Is it worth giving up global sourcing and production? |
| | Community Learning | • Can your project support the local community to inform itself of environmental and social issues surrounding clothing? |
| | Creative Enterprise | • Can your product and service help to create social enterprise and benefit society and the environment? |
| | Power Shift | • Can your design help to create new forms of cultural exchange and support socially marginalized people? |
| | Social Feedback | • How can your product service systems be catalyzed by active social feedback and learning processes? |
| | Way of Living | • How does your design support people to lead a greener, healthier, and happier lifestyle? |

## Appendix B. Open-Ended Questions: Empowerment, Persuasion, and Interaction Patterns

| | | |
|---|---|---|
| **Empowerment** | Storytelling | • How can you make multiple layers of storytelling through a series of discoveries that enhance emotional connections between the product/service system and users? |
| | Magic | • Can your design evoke a magical experience and curiosity that creates emotional attachments in the users? |
| | Poetic | • Can your design trigger memory as a poetic experience and evoke personal intimacy? |
| | Playfulness | • How can you maximize playful experiences through multisensory design that maximizes emotional and social values? |
| | Personality | • Can your design offer different options that depend on the user's personality or characters that provoke emotional attachment and social enjoyment? |
| | Partial Completion | • Can your design be rearranged in various configurations or semifinished platforms so that other users can participant in the design process? |
| | User as Maker | • How can your design encourage users to become makers and facilitate co-creative sustainable consumption? |
| | Smart Craft | • Can your design cultivate craft skills in which the user can easily learn the traditional craft with a combination of new technologies and science? |
| | Open-source Fashion | • Can your design utilize open-source innovation through socially engaged processes with a wide range of social groups? |
| | Cultivating Creativities | • Can your design system allow everyday people a way to become more creative and get inspired to create using discarded materials? |
| **Persuasion** | Informative | • How does your design enhance more informed decisions for both clothing producers and consumers? |
| | Guidance | • Can your products or services entail the appropriate guidance to the user to promote pro-environmental behaviors? |
| | Story of Use | • Can your design scenario promote a more sustainable way of living in the context of their everyday behavior as a story? |
| | Transparency | • Can you make your design more transparent by revealing what is under the surface of the design process? Can this be used to influence the user's perceptions and behavior? |
| | Warning | • Can your design reflect the user's energy consumption or carbon footprints depending on their behavior? |
| | Reinforcement | • Can you emphasize the potential of new energy sources and increase awareness of the value of resource use through your design? |
| | Reward | • Can you encourage the user to participate in continuous positive action through a series of rewards or incentives when they achieve positive actions in your system? |
| | Simplicity | • Can you simplify your design system or service to be easier, so that people's behavior can adapt to habitual routine? |
| | Commitment | • How can your design create a dialogue with the user that encourages responsible use of products? How could we actively promote garments whose environmental performance is reliable? |
| | Shareholder Incentive | • Can your design promote continuous improvement of shareholders' values? |
| **Interaction** | Sensory Effect | • How can the use of sensory effects (sound, touch, light, vision, smell, taste) assist users to interact [with] more pro-environmental behavior? |
| | Parameter Change | • How can you support users to change negative habits, without controlling user behavior automatically, through design combined with advanced technology? |
| | Preliminary Action | • Can your design protect from danger or harmful environmental impacts? |
| | Reactive Fashion | • Can your design automatically respond to user behavior and outside environmental conditions such as light, movement, touch, etc.? |
| | Fairytale Fashion | • Can your design provoke curiosity and multiple discoveries through various interactions with garments or your design system? |
| | Tailoring | • How can your design meet individual user needs as well as supporting production efficiency? |
| | Notification | • Can your design service and system allow users to obtain just-in-time garment information? |
| | Feed Forward | • Can your user obtain a preview of future scenarios of products or services that demonstrate the consequences of their different actions or choices? |
| | Behavior Feedback | • How can your design help users to motivate pro-environmental behavioral [change] through providing reflective consumer feedback? |
| | Environmental Response | • Can your design communicate with people to encourage them to practice pro-environmental behavior through the use of environmental response? |

## Appendix C. A Summary of Participants' Perspectives on Each Card Pattern

| Patterns | Sample responses |
|---|---|
| Choice | *"Looking at different ways of sustainability"* |
| | *"Interesting got me thinking," "To learn how it affects society"* |
| | *"It seemed more relevant to look at a particular task"* |
| Optimisation | *"It is a more active approach" "Interesting to see how you can maintain products" "It talks about essential topic of sustainability and practices which need to be implemented"* |
| Social Conversation | *"Something more involved with a modern future" "It's useful due to the social responsibilities and importance"* |
| | *"It is interesting to see how solutions can be generated in a social way" "It is most fun"* |
| Empowerment | *"It shows how you can personalize products to suit consumer needs"* |
| | *"It is [an] alternative possible sustainable approach"* |
| | *"It was most interesting"* |
| Interaction | *"It had scope for future development"* |
| | *"I like the idea of incorporating interacting into sustainable product and services" "The tool provides a lot of interaction solutions which expand the designers' horizon"* |
| Persuasion | *"It is good to know other ideas and rewarding to reach a conclusion and solution as a group"* |
| | *"To persuade other partners to accept your idea is quite important" "Awareness is important"* |
| Other opinions | *"All equally same value" "All very well understood"* |
| | *"Because of the cards, we had already set down paths of ideas"* |
| | *"The different sections of the toolkit helped generate ideas"* |
| | *"Good to see questions, the cards, that make you think more about your design"* |
| | *"Different ideas are merged to get a unique idea"* |
| | *"Working on a new concept logically really helps to bring out lot of things-innovation at its best"* |

## Appendix D. Workshops: Idea Evaluation and Critical Reflection

**What are the key values or impacts of your design?**
In a group, discuss whether your concept is addressing feasible and useful for environmental, social and economic sustainability and other design values.

| Finalise the solution | | Examples of critical evaluation | Score – Impact of sustainability: 1 (low) to 10 (high) |
|---|---|---|---|
| Environmental value | ✓ | Resource usage rates (e.g., water, chemical inputs, fibres, textiles, other materials) | |
| | ✓ | Optimisation of sustainable consumption rate (e.g., reuse, recycle and upcycle) | 1---2---3---4---5---6---7---8---9---10 |
| Social and ethical value | ✓ | Textile and garment workers' well-being and wages | 1---2---3---4---5---6---7---8---9---10 |
| | | | 1---2---3---4---5---6---7---8---9---10 |
| | ✓ | Job creation | 1---2---3---4---5---6---7---8---9---10 |
| | ✓ | Positive social engagement | 1---2---3---4---5---6---7---8---9---10 |
| | ✓ | Health and safety improvements | 1---2---3---4---5---6---7---8---9---10 |
| | ✓ | Animal welfare | |
| Economic value | ✓ | Financial value for the company and shareholders | 1---2---3---4---5---6---7---8---9---10 |
| | ✓ | Competitive pricing and commercial feasibility of the project | 1---2---3---4---5---6---7---8---9---10 |
| Novelty value | ✓ | Unique and original aspects of your design | 1---2---3---4---5---6---7---8---9---10 |
| Other design values | ✓ | Maximisation of aesthetic potential | 1---2---3---4---5---6---7---8---9---10 |
| | ✓ | Experiential and symbolic values | 1---2---3---4---5---6---7---8---9---10 |
| | ✓ | Self-expressive values | 1---2---3---4---5---6---7---8---9---10 |

**Appendix E. Examples of User-Generated Concepts from Workshops**

| Group | A | B |
|---|---|---|
| Idea visualisation |  |  |
| Target market | Babies and parents | Online and offline shoppers |
| Identified issues and needs | Babies grow very quickly and their parents need to buy new garments too often | Lack of personalisation and personal experience in conventional shopping stories |
| Suggested design solutions | New product design and *product–service systems (PSS)* for babies. To solve this problem, after participants selected the alternative material and modularity cards, they suggested designs for baby clothing by observing users' behaviours. The garments can be detached or separated for different purposes and, because they used stretch fabric, when a baby grows, the garment grows with them. *PSS* allows parents to share and buy second-hand children's products instead of buying new ones that require only short-term use. | Retail shops can cater to individual users' needs by changing the ways users' shop, providing alternative, higher quality, and smart fabric selections. Incorporating the idea of smart DIY fashion, consumers could buy fabrics and colour swatches with video instructions. The consumer could then design and produce the garment at home, emphasising self-sufficiency and effective communication with consumers. The manufacturing process would need to be more transparent; fashion companies sell various services, including knowledge, ideas, and production qualities. Production and consumption processes would be continually improved based on an effective consumer feedback loop. |
| Group | C | D |
| Idea visualisation |  |  |
| Target market | Local communities | Local communities |
| Identified issues and needs | Lack of accessibility of personalised design in local community | Lack of local clothing-sharing services and communities |
| Suggested design solutions | This group considered that the way to achieve sustainable fashion is through good quality design. Participants selected the 'tailoring', 'user as a maker', and 'shareholder incentive' cards. Participants suggested a community project that collaborates between a tailor/designer and a local or supermarket retailer by bringing expertise to an assessable level. A tailor/designer-run workshop for the local community would foster more personalised styles and looks using high-quality and sustainable materials. | An online platform for a global swap and share service with local communities. The clothing library can be activated from community to community and sell the intangible value of the services. Furthermore, various users can wear versatile, one-size clothing without size limitations. |

|  | E | F |
|---|---|---|
| Idea visualisation | 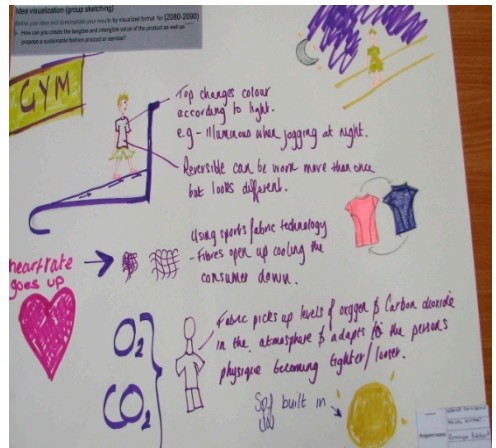 | 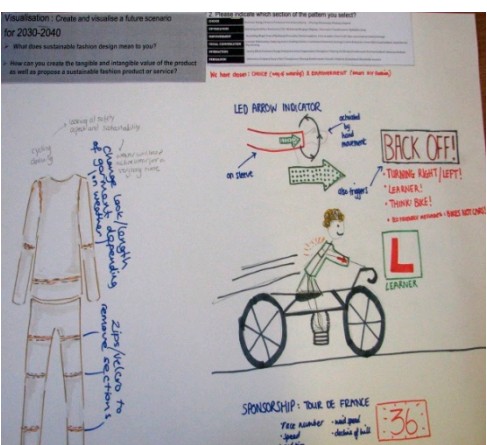 |
| Target market | Consumers who are concerned about wellness and sustainability | Future bicyclists |
| Identified issues and needs | Lack of function in clothing (safety, comfort, utility) and consumers' emotional and behavioural interactions with products | Lack of flexibility, comfort, and safety elements in clothing for night-time bicyclists |
| Suggested design solutions | Garments can offer protection from pollution and react to the climate, such as wind, sun, and rain. Adaptable, UV-resistant, waterproof, breathable fabric has reacting fibres that can expand and contract responding to the temperature and weather. The clothing also reacts depending on the wearer's movement and temperature and how they are breathing. Garments can react to different senses (smell, sound, touch). When people are hugging, the garments can interact with the wearers to enhance a couple's relationship. | To encourage safe motorcycle or bicycle riding, the back of the LED jacket gives an indication of the wearer's level of cycling skills, handle movement, and speed. Incorporating the idea of design modularity and the use of zippers or Velcro, clothing can be changed; the look and length of the garment and its sections can be removed, depending on the user's activities and temperature. The design components can be interchanged with other functions and adjusted to different sizes. The fashion company could sell sections of the modular components and provide sponsorships for safe riders. |

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
