# Peer review of "Fostering Sustainable Fashion Innovation: Insights from Ideation Tool Development and Co-Creation Workshops"

_sustainability, doi:10.3390/su152115499_

Round 1
Reviewer 1 Report
Comments and Suggestions for Authors
Thank you for the opportunity to revise this paper, titled: Fostering Sustainable Fashion Innovation: Insights from Ideation Tool Development and Co-Creation Workshops.
First of all, congratulations to the authors.
The proposal made is very actual and interesting, and the sector studied is important and relevant to the fashion economy of Europe and the world.
In general, small errors are detected, and in essence, it is a literature review and a toolkit proposal to increase the sustainability of the fashion sector, illustrating it with examples of concrete applications.
It is only necessary to indicate some minor revision:
- Literature review. The content can be interesting, but it must be proposed appropriately. I suggest to present the literature review in a proper analytic manner.
- Discussion. This section must present your findings – the findings and what we can learn from your work.
- The conclusions could be stronger. What can a designer take from your paper? The limitations and future research perspectives could be better and more developed.
- Also I suggest checking the document for typos/editing required.
I hope that the author(s) would sincerely find my comments constructive to improve the scientific orientation of the article before considering submission to some other journal.
All the best.
Comments on the Quality of English LanguageMinor spelling revisions are needed.
Author Response
Thank you very much for taking the time to review this manuscript and offering constructive suggestions. Please find the detailed responses below, and the corresponding revisions are highlighted in yellow in the resubmitted file.
Yes |
Can be improved |
Must be improved |
Not applicable |
|
Is the content succinctly described and contextualized with respect to previous and present theoretical background and empirical research (if applicable) on the topic? |
(x) |
( ) |
( ) |
( ) |
Are all the cited references relevant to the research? |
(x) |
( ) |
( ) |
( ) |
Are the research design, questions, hypotheses and methods clearly stated? |
(x) |
( ) |
( ) |
( ) |
Are the arguments and discussion of findings coherent, balanced and compelling? |
( ) |
(x) |
( ) |
( ) |
For empirical research, are the results clearly presented? |
(x) |
( ) |
( ) |
( ) |
Is the article adequately referenced? |
(x) |
( ) |
( ) |
( ) |
Are the conclusions thoroughly supported by the results presented in the article or referenced in secondary literature? |
(x) |
( ) |
( ) |
( ) |
Literature review. The content can be interesting, but it must be proposed appropriately. I suggest to present the literature review in a proper analytic manner.
- I appreciate your comment on the literature review. In response, we have included additional literature, particularly by providing a clear definition of sustainable fashion innovation. We have critically analysed existing sustainable innovation tools, considering their applicability at four different levels of sustainable innovations. We've also discussed the limitations and advantages of their usage.
- Discussion. This section must present your findings – the findings and what we can learn from your work.
Could be presented as more scientific orientation of the article
- Thank you for emphasising the importance of the discussion section. In line with your suggestion, we've presented key themes derived from our research findings in this section. We've delved deeper into the implications of these findings, drawing from insights gained during interviews and workshops. We're also working to present the article with a more scientific orientation.
- The conclusions could be stronger. What can a designer take from your paper? The limitations and future research perspectives could be better and more developed.
- We've enhanced the paper by summarising the contribution of our research findings, making it clearer for designers to extract actionable insights. We've also expanded on the limitations and provided more detailed perspectives for future research in the discussion section.
Checking the document for typos/editing required.
- We've diligently reviewed the document, addressing issues related to referencing, typos, and the formatting of tables, figures, and appendices.
We hope that you are satisfied with these changes. Thank you for reviewing this article.

Reviewer 2 Report
Comments and Suggestions for Authors
The work with the title "Fostering Sustainable Fashion Innovation: Insights from Ideation Tool Development and Co-Creation Workshops" is at the interface between art and science (technology).
As the authors claim, they “proposed and tested the application of a sustainable fashion tool (SFB) in the co-design workshop setting, reviewed how effectively it meets question 4’s criteria and considered. Finally,they considered whether SFB has ‘stood the test of time’ as new sustainability issues have emerged and made recommendations for its future development. This study contributes to the development of a sustainable fashion innovation framework for the co-design workshop and supports the creation of novel product-service system design that aligns with sustainable fashion innovations”.
I believe that the objectives of the work (which the authors themselves proposed) were hardly achieved. The work is rather a beginning of research. However, due to the fact that the paper refers to the field of sustainability and innovation, the paper deserves to be published in the MDPI revue.
Author Response
Thank you very much for taking the time to review this manuscript and offering constructive suggestions. Please find the detailed responses below, and the corresponding revisions are highlighted in yellow in the resubmitted file.
Yes |
Can be improved |
Must be improved |
Not applicable |
|
Is the content succinctly described and contextualized with respect to previous and present theoretical background and empirical research (if applicable) on the topic? |
(x) |
( ) |
( ) |
( ) |
Are all the cited references relevant to the research? |
( ) |
(x) |
( ) |
( ) |
Are the research design, questions, hypotheses and methods clearly stated? |
( ) |
(x) |
( ) |
( ) |
Are the arguments and discussion of findings coherent, balanced and compelling? |
( ) |
(x) |
( ) |
( ) |
For empirical research, are the results clearly presented? |
( ) |
(x) |
( ) |
( ) |
Is the article adequately referenced? |
(x) |
( ) |
( ) |
( ) |
Are the conclusions thoroughly supported by the results presented in the article or referenced in secondary literature? |
(x) |
( ) |
( ) |
( ) |
Comments and Suggestions for Authors
The work with the title "Fostering Sustainable Fashion Innovation: Insights from Ideation Tool Development and Co-Creation Workshops" is at the interface between art and science (technology).
As the authors claim, they “proposed and tested the application of a sustainable fashion tool (SFB) in the co-design workshop setting, reviewed how effectively it meets question 4’s criteria and considered. Finally, they considered whether SFB has ‘stood the test of time’ as new sustainability issues have emerged and made recommendations for its future development. This study contributes to the development of a sustainable fashion innovation framework for the co-design workshop and supports the creation of novel product-service system design that aligns with sustainable fashion innovations”.
I believe that the objectives of the work (which the authors themselves proposed) were hardly achieved. The work is rather a beginning of research. However, due to the fact that the paper refers to the field of sustainability and innovation, the paper deserves to be published in the MDPI revue.
- Thank you for taking the time to review this article and for your valuable suggestions. Your feedback prompted a further re-evaluation of our research objectives and findings. We have reviewed the research questions and objectives to ensure that we have articulated them effectively in the context of empirical research findings from workshops and interviews. The discussion section now delves deeper into the implications of our co-creation workshop findings, particularly regarding how different categories of co-creation users can effectively leverage the toolkit, as well as strategies for classifying diverse co-creation participants. We have also included specific examples of co-creation workshop outputs, featuring concepts generated by the participants.
To enhance the clarity of our work, we have provided a more extensive clarification of the key contributions to knowledge and practical implications of our study. This should help readers better grasp the value and potential impacts of our paper.It is important to regard this paper as an initial step in a broader research journey. While our findings may not be definitive, they serve as a foundation for further exploration. The paper retains its relevance within the fields of sustainability and innovation.
We appreciate your comments, and in response, we have conducted a thorough review of additional literature pertaining to the definition of sustainable fashion innovation and a critical analysis of existing tools in the literature review section. Furthermore, we have highlighted our contributions to knowledge in the conclusion section. The discussion section now prominently addresses the limitations of our study, outlines future research directions, and offers a practical perspective to guide fellow researchers in this field.
- To enhance the clarity of our work, we have provided a more extensive clarification of the key contributions to knowledge and practical implications of our study. This should help readers better grasp the value and potential impacts of our paper. While our findings may not be definitive, they serve as a foundation for further exploration. The paper retains its relevance within the fields of sustainability and innovation. We appreciate your comments, and in response, we have conducted a thorough review of additional literature pertaining to the definition of sustainable fashion innovation and a critical analysis of existing tools in the literature review section. Furthermore, we have highlighted our contributions to knowledge in the conclusion section. The discussion section now prominently addresses the limitations of our study, outlines future research directions, and offers a practical perspective to guide fellow researchers in this field.
We hope that you are satisfied with these changes. Thank you for reviewing this article.

Reviewer 3 Report
Comments and Suggestions for Authors
This article aims to promote sustainable fashion innovation through the development of ideation tool, and I believe there are significant shortcomings in the following aspects:
1. The innovation of this study is not sufficient. The title of this article is "Fostering Sustainable Fashion Innovation: Insights from Ideation Tool Development and Co-Creation Workshops", but the theoretical innovation of the research topic in the article is not significant.
2. The research method is not perfect enough. The Sustainable Fashion Bridges (SFB) tool has been applied in this paper, but whether this tool can be combined with “Workshops” remains to be explored.
3.In “Research Methods”, the participants are divided into 8 groups. Is the homogeneity of each group's members conducive to creativity.
4. The results section lacks quantitative analysis.
5.In P6, Line259, "Figure 2" is mentioned , but there is no Figure 2 in the manuscript.
Comments on the Quality of English LanguageMinor editing of English language required.
Author Response
Thank you very much for taking the time to review this manuscript and offering constructive suggestions. Please find the detailed responses below, and the corresponding revisions are highlighted in yellow in the resubmitted file.
Yes |
Can be improved |
Must be improved |
Not applicable |
|
Is the content succinctly described and contextualized with respect to previous and present theoretical background and empirical research (if applicable) on the topic? |
( ) |
( ) |
(x) |
( ) |
Are all the cited references relevant to the research? |
( ) |
( ) |
( ) |
(x) |
Are the research design, questions, hypotheses and methods clearly stated? |
( ) |
( ) |
(x) |
( ) |
Are the arguments and discussion of findings coherent, balanced and compelling? |
( ) |
( ) |
(x) |
( ) |
For empirical research, are the results clearly presented? |
( ) |
( ) |
(x) |
( ) |
Is the article adequately referenced? |
( ) |
( ) |
(x) |
( ) |
Are the conclusions thoroughly supported by the results presented in the article or referenced in secondary literature? |
( ) |
( ) |
(x) |
( ) |
Comments and Suggestions for Authors
This article aims to promote sustainable fashion innovation through the development of ideation tool, and I believe there are significant shortcomings in the following aspects:
- The innovation of this study is not sufficient. The title of this article is "Fostering Sustainable Fashion Innovation: Insights from Ideation Tool Development and Co-Creation Workshops", but the theoretical innovation of the research topic in the article is not significant.
- Response: We've taken your feedback and made substantial improvements. The theoretical background, including the definition of sustainable innovation and a critical examination of existing tools, has been thoroughly reviewed. In the discussion section, we have explored how different categories of co-design users can effectively utilise the sustainable fashion innovation tool. Furthermore, we've defined the sustainability skills necessary for professionals in the sustainable fashion industry, based on in-depth interviews with industry professionals and educators.
- The research method is not perfect enough. The Sustainable Fashion Bridges (SFB) tool has been applied in this paper, but whether this tool can be combined with “Workshops” remains to be explored.
- Response: We acknowledge that our research method has its limitations. The participants for the workshops were selected through convenience sampling. As a result, we have included discussions of these limitations and offered recommendations for further studies that could advance the theory and practice of sustainable fashion innovation research.
3.In “Research Methods”, the participants are divided into 8 groups. Is the homogeneity of each group's members conducive to creativity.
- Response: While some groups were relatively homogeneous, such as workshop group 8, which included both industry professionals and students, we recognise the importance of diversifying participant backgrounds. We've suggested future research exploring co-creation workshops with a more diverse range of participants to address this concern.
- The results section lacks quantitative analysis.
- Response: This study predominantly relies on qualitative methods, involving workshops and in-depth interviews. However, we acknowledge the need for more robust quantitative data analysis in future studies, especially involving different types of co-creation groups.
5.In P6, Line259, "Figure 2" is mentioned , but there is no Figure 2 in the manuscript.
- Response: We've conducted a thorough review of the manuscript, including figures, tables, and appendices, to identify and rectify any formatting errors or missing information. Your attention to detail is much appreciated.
We hope that you are satisfied with these changes. Thank you for reviewing this article.

Round 2
Reviewer 3 Report
Comments and Suggestions for Authors
I think the authors have made revisions to address and respond to the issues and questions raised in the previous review. The manuscript has been improved.
Comments on the Quality of English LanguageMinor editing of English language required.
Author Response
Dear editors,
I hope this letter finds you well. I am writing in response to the comments provided by the reviewer on our manuscript. We appreciate the reviewer's constructive feedback. We have performed proofreading and editing on the manuscript to enhance its overall readability. If any further editing is necessary, please do not hesitate to inform us, and we will promptly address any remaining issues or further improve the quality of the paper. Thank you once again for considering our manuscript.
Sincerely,
Eunsuk
Dr Eunsuk Hur FHEA
1.66 Clothworkers' Building Central | School of Design | University of Leeds
E: e.s.hur@leeds.ac.uk
